# Prioritization of Fluorescence In Situ Hybridization (FISH) Probes for Differentiating Primary Sites of Neuroendocrine Tumors with Machine Learning

**DOI:** 10.3390/ijms242417401

**Published:** 2023-12-12

**Authors:** Lucas Pietan, Hayley Vaughn, James R. Howe, Andrew M. Bellizzi, Brian J. Smith, Benjamin Darbro, Terry Braun, Thomas Casavant

**Affiliations:** 1Interdisciplinary Graduate Program in Genetics, University of Iowa, Iowa City, IA 52242, USA; lucas-pietan@uiowa.edu (L.P.); hayley-vaughn@uiowa.edu (H.V.); terry-braun@uiowa.edu (T.B.); 2Department of Biomedical Engineering, University of Iowa, Iowa City, IA 52242, USA; 3Stead Family Department of Pediatrics, University of Iowa, Iowa City, IA 52242, USA; 4Healthcare Department of Surgery, University of Iowa, Iowa City, IA 52242, USA; james-howe@uiowa.edu; 5Department of Pathology, University of Iowa, Iowa City, IA 52242, USA; andrew-bellizzi@uiowa.edu; 6Department of Biostatistics, University of Iowa, Iowa City, IA 52242, USA; brian-j-smith@uiowa.edu; 7Center for Bioinformatics and Computational Biology, University of Iowa, Iowa City, IA 52242, USA; 8Department of Electrical and Computer Engineering, University of Iowa, Iowa City, IA 52242, USA

**Keywords:** fluorescence in situ hybridization, neuroendocrine tumor, machine learning, biomarker, imputation, model

## Abstract

Determining neuroendocrine tumor (NET) primary sites is pivotal for patient care as pancreatic NETs (pNETs) and small bowel NETs (sbNETs) have distinct treatment approaches. The diagnostic power and prioritization of fluorescence in situ hybridization (FISH) assay biomarkers for establishing primary sites has not been thoroughly investigated using machine learning (ML) techniques. We trained ML models on FISH assay metrics from 85 sbNET and 59 pNET samples for primary site prediction. Exploring multiple methods for imputing missing data, the impute-by-median dataset coupled with a support vector machine model achieved the highest classification accuracy of 93.1% on a held-out test set, with the top importance variables originating from the *ERBB2* FISH probe. Due to the greater interpretability of decision tree (DT) models, we fit DT models to ten dataset splits, achieving optimal performance with k-nearest neighbor (KNN) imputed data and a transformation to single categorical biomarker probe variables, with a mean accuracy of 81.4%, on held-out test sets. *ERBB2* and *MET* variables ranked as top-performing features in 9 of 10 DT models and the full dataset model. These findings offer probabilistic guidance for FISH testing, emphasizing the prioritization of the *ERBB2*, *SMAD4*, and *CDKN2A* FISH probes in diagnosing NET primary sites.

## 1. Introduction

Neuroendocrine tumors (NETs) are a diverse collection of tumors arising from neuroendocrine cells, which are present in most bodily tissues [1]. The latest data reveals a rise in the incidence and the prevalence of NETs, with a prevalence of just over 170,000 reported cases in the United States [1,2]. Gastroenteropancreatic NETs are the most common NETs, with small bowel NETs (sbNETs) accounting for 17–20% and pancreatic NETs (pNETs) comprising 7–10% of all NET cases [3,4]. NETs can present either asymptomatically or with a range of symptoms that may resemble other medical conditions, potentially causing delays in diagnosis and leading to more advanced stages of cancer [5,6,7]. Metastasis is common with sbNETs and pNETs, most often in the liver, where many patients present with an unknown primary site [8]. Determining the primary site of NETs is pivotal for patient care as treatment approaches differ for sbNETs and pNETs and the primary site has important implications for surgical management [9].

Imaging continues to hold a crucial role for diagnosis and establishing NET primary sites, although it yields inconclusive results in 12–22% of cases [10,11,12]. Blood peptide and protein biomarkers have been identified for diagnosing primary sites with limited validation and results [13,14]. However, histological classification with immunohistochemistry (IHC) persists as the primary approach for diagnosis, with no universally standardized diagnostic algorithm in place [15,16]. Several IHC biomarkers have been established for NET diagnosis. At the University of Iowa, the clinical classification model for distinguishing NET subtypes using IHC biomarkers has a 90% sensitivity for sbNETs and pNETs [15]. Molecular tests have the potential to be extremely useful in determining NET primary sites. Recently, a discovery and validation study found association of eight fluorescence in situ hybridization (FISH) test probes with sbNET and pNET primary sites [17]. FISH tests assess biopsied NET cells for copy number alterations in gene biomarkers, providing a count of cells exhibiting either a loss in, normal, or gain in number of copies of the gene. Clinically validated thresholds for these counts are applied to determine the final test results.

Machine learning (ML) models offer the advantage of concurrently assessing multiple features, discovering patterns and interactions in the data undetected by human observation or conventional analytical methods. ML methods have found widespread application in healthcare research for predicting diagnoses, treatment options, and prognoses, yielding clinically implementable results and impacting patient care [18]. In cancer research, ML has been used to explore the development of new clinical biomarker tests and augment the detection capabilities of established tests and imaging methods [19]. ML methods have been applied to classify sbNET and pNET samples from multiple blood protein biomarkers, with results contributing to the recent push to establish new biomarkers or a multianalyte test for NETs [20,21]. In a similar effort, ML models fitted with microRNA (miRNA) marker expression data have had success in classifying NET subtypes [22]. The NETest is another multianalyte analysis of circulating transcripts that employs ML algorithms to predict NETs compared to controls with a classification accuracy of 94% for small intestinal NETs and 91% for pNETs [23,24,25]. However, concerns regarding cost-effectiveness and accessibility for the NETest have been raised [13].

Feature importance can be assessed across ML models, identifying the variables that are significantly influencing predictive performance and are contributing the most important information [26]. While many ML models lack interpretability for how predictions are generated, decision tree (DT) models offer transparency with the ability of visualizing the model as a series of decisions for predictions [27].

In this study, we investigated eight FISH biomarker probes and assessed the resulting metrics with ML techniques for the prediction of sbNET and pNET primary sites with the main aim of prioritizing FISH probes for diagnosis. To accomplish this, we assessed variable importance for predictions from the best performing models and used DT models for plotting algorithmic prioritization of FISH probes. Our findings suggest prioritization of *ERBB2*, *SMAD4*, and *CDKN2A* FISH probes for determination of sbNET and pNET primary sites.

## 2. Results

A total of 8 FISH test gene biomarker probes were assessed for 144 patient samples consisting of 85 sbNETs (59%) and 59 pNETs (41%). Each sample had an average of 3.13 gene biomarker FISH test results with 5 samples having only 1 gene biomarker test and 6 samples with 7 gene biomarker tests (Table 1). No sample had all 8 gene biomarker tests performed. Among the 8 FISH probes, the *MET* test was performed on 93 samples, while the *CDKN2A* test was conducted on 36 samples (Table 2).

Implementation of the naïve bayes (NB) and extreme gradient boosted tree (XGBTree) models were able to be fit to the training set with missing data. The naïve bayes model outperformed the XGBTree model on the training set with 10-fold cross-validation (CV) mean brier score = 0.156 (lower is better), mean accuracy = 83.5%, and mean area under the receiver operating characteristic (AUROC) curve = 0.869, as well as on the held-out test set with brier score = 0.117, accuracy = 86.2%, and AUROC curve = 0.928 (Table 3 and Appendix A). Top models with parameters are included in Appendix A. Relative permutation-based variable importance (PVI) for the naïve bayes model resulted in an *ERBB2* biomarker variable being most important (Figure 1). To address the missing data, imputation methods were employed, and all models were fitted to the training set. Substituting missing data with variable medians with the support vector machine with radial basis kernel function (SVM-RB) model yielded the best performance on the training set: mean brier score = 0.115, mean accuracy = 85.4%, mean AUROC curve = 0.919. On the held-out test set, the SVM-RB model had a predictive performance of brier score = 0.0658, accuracy = 93.1%, and AUROC curve = 0.861 (Table 3, Figure 2A, and Appendix A). The top two variables for the SVM-RB model are ERBB2_loss and ERBB2_gain (Figure 2B). The decision tree model, trained on the median-imputed dataset, achieved a mean training accuracy of 75.5% and a testing accuracy of 72.4%, with the first four decision tree splits based on *ERBB2* and *CCNE1* variables (Table 3 and Figure 3A). The top two variables for the decision tree model by PVI include ERBB2_gain and CCNE1_gain (Figure 3B).

Cytogenetic standard thresholds for FISH test results when considering a loss, normal, and gain were applied to the initial dataset after imputation to construct single categorical variables for each gene biomarker. Models fitted to the bagged trees imputed dataset performed slightly worse than the datasets imputed by other methods (Appendix A). Imputation by mean, median, and k-nearest neighbor (KNN) performed similarly across most models with the random forest and SVM-RB models performing slightly better on the training and test sets (Table 4, Appendix A). The random forest model performed best on the mean-imputed training and test sets, having a training mean brier score = 0.174, mean accuracy = 77.2%, and mean AUROC curve = 0.797, and a testing brier score = 0.147, accuracy = 75.9, and AUROC curve = 0.885 (Table 4, Appendix A). SVM-RB models excelled on median-imputed and KNN-imputed training data, with the KNN dataset showing the best performance: mean brier score = 0.168, mean accuracy = 79.1%, and mean AUROC curve = 0.786 (Table 4, Appendix A). On the testing set, the SVM-RB models had an increase in performance for the median-imputed and KNN-imputed datasets with the best performance on the median dataset: brier score = 0.158, accuracy = 79.3%, and AUROC curve = 0.916 (Table 4, Appendix A). The best test performance on the median-imputed dataset was with the random forest model with brier score = 0.133, accuracy = 79.3%, and AUROC curve = 0.930 (Table 4, Appendix A). For the KNN-imputed test set, the random forest model had the best brier score = 0.154 and AUROC curve = 0.873, while the decision tree model had the best accuracy = 82.8% (Table 4, Appendix A).

The DT model performed best on the KNN-imputed dataset with a transformation to single categorical biomarker variables. Ten random training-testing splits were constructed for Monte Carlo CV and sample summary statistics for the split datasets and probabilities for a given FISH test result for the KNN dataset are given in the Appendix A (Appendix A). A DT model was fitted to each training split with average estimated predictive performance on all splits having a mean brier score of 0.185 (0.161–0.228), mean accuracy of 75.9% (range of 63.5–80.4%), and a mean AUROC curve of 0.764 (0.696–0.827) (Table 5). Test set performance of the ten models had a mean accuracy of 81.4% (range of 69.0–89.7%), a mean brier score of 0.156 (0.104–0.252), and a mean AUROC curve of 0.816 (0.710–0.909) (Table 5). In 8 of 10 tree models, the *ERBB2* variable was selected as the initial split of the DT, with *ERBB2* gain (3) samples proceeding down the left branch and *ERBB2* loss and normal (1 and 2) samples proceeding down the right branch of the trees (Figure 4, Appendix A). The *ERBB2* variable was included as a decision split in 9 tree models and selected as a split for 11 of the 49 total splits across all trees (Appendix A). The *SMAD4* variables were the second most featured for 7 of the 49 total splits across all trees (Appendix A). Three tree models incorporated only a single split with two utilizing the *ERBB2* variable and the third utilizing the *CDKN2A* variable (Appendix A, Appendix A). For all tree models, 89.9% of the branch of decisions for classification at the terminal nodes have a probability greater than a majority class classifier for each split (Appendix A). PVI yielded *ERBB2* and *MET* biomarker variables at the first and second positions for 9 of the 10 decision tree models. *ERBB2* and *MET* biomarker variables consisted of the top five variables of importance for 6 of 10 models (Figure 4C, Appendix A). The consensus tree model generated from the predictive probabilities of all ten DT models has a brier score = 0.135, accuracy = 79.2%, and AUROC curve = 0.908 (Appendix A, Appendix A). The distribution of correctly classified samples by all ten DT models is close to expected, with 82 samples (27 pNETs, 55 sbNETs) being correctly classified by all ten models and 21 samples (7 pNETs, 14 sbNETs) being correctly classified by 9 of the 10 models. There were 9 samples (8 pNETs, 1 sbNETs) misclassified by all ten models, 7 samples (6 pNETs, 1 sbNETs) misclassified by nine models, and 9 samples (4 pNETs, 5 sbNETs) misclassified by eight models (Appendix A).

The DT model trained on the full dataset (all 144 samples) generated tree splitting on three variables. The first split is on the *ERBB2* variable, followed by splits on the *CDKN2*A and *SMAD4* variables (Figure 5A). Estimated predictive performance with 10-fold CV yielded a mean brier score = 0.161, mean accuracy = 79.8%, and mean AUROC curve = 0.800. The tree performance was assessed with the same dataset used to train the model and had a brier score = 0.158, an accuracy = 79.2%, and an AUROC curve = 0.793 (Figure 5B, Appendix A). Three of the four terminal nodes for the full dataset model have a prediction probability greater than the majority class classifier at 0.590 (Appendix A). PVI resulted with the top two variables being *ERBB2* gain and *MET* gain (Figure 5C).

## 3. Discussion

In this study, we investigated the predictive capabilities of results obtained from eight FISH gene biomarker probes for the classification of sbNETs and pNETs. The primary aim was to prioritize FISH probes for the diagnosis of primary sites in patients presenting with suspected sbNETs or pNETs. To our knowledge, this is the first study to attempt to establish an algorithmic standard for the prioritization of FISH tests in NETs diagnostics using machine learning techniques. To achieve this, we subjected numerous variations of FISH result data to ML analyses to classify sbNET and pNET samples. Our rationale behind this approach was that ML models can leverage multiple probes collectively for prediction. The approach enabled us to discern the influence of individual probes on predictions, facilitating prioritization. FISH tests offer clear advantages over other proposed multi-biomarker (multianalyte) tests. They are common histological tests in cancer fields and are frequently used to provide confirmation in cases where immunohistochemistry and imaging tests yield unclear results [14,15,16]. FISH is a widely available clinical test, offering a cost-effective approach compared to more elaborate methods and has a relatively swift turnaround time for results. Through the prioritization of FISH probes in this study, there is a potential to improve the diagnostic assessment of NETs and reduce the cost and the time of results, improving patient care. The eight FISH probes selected to include in our analysis underwent prior examination in a discovery and validation study, revealing individual associations with sbNETs and/or pNETs [17]. Gene probes were chosen for their commercial availability, targeting specific genomic regions associated with NET subtypes through copy number alteration analysis [17]. Our analysis prioritizes probes for diagnosing NET primary sites but does not offer evidence for selected genes and the pathogenesis of tumors. Common copy number alterations of whole chromosomes and chromosome regions have been previously found in sbNETs and pNETs [28,29]. The data included in the present study were generated for the validation analysis [17]. Our top-performing models can discern sbNET from pNET samples across all performance metrics. Assessing variable importance in our top models, coupled with decision tree plots, enables us to prioritize FISH probes, allowing for the establishment of algorithmic guidelines for testing. 

The complete sample set consisted of 59% sbNETs and 41% pNETs, comparable to estimates of prevalence percentages in the population. The inclusion of a higher number of pNET cases allows us to better identify patterns within pNETs, aligning with our objective of developing a broadly applicable and generalizable model for sbNETs and pNETs. Imputation of missing data enabled all models to be fitted to the dataset and assessed for performance, in contrast to only two models accommodating the extent of missing data present in the raw dataset. We conducted multiple iterations of transformations of the dataset, incorporating biological thresholds, to identify the optimal dataset for distinguishing sbNETs from pNETs. Among all the datasets and models we examined, imputation by mean, median, and KNN demonstrated comparable performances, whereas datasets imputed by bagged tree models generally exhibited weaker results. In general, SVM and random forest (RF) models consistently outperformed other models across all datasets.

The development of new biomarkers for diagnosing NET subtypes has been a long-standing research focus in the field. While single biomarker tests have proven effective in identifying certain NET subtypes, they are uninformative for others, leading to recent increased efforts to establish a multi-biomarker strategy [10,13]. Our multi-probe analysis falls within this category, with a primary focus on prioritizing FISH probes for the classification of sbNET and pNET primary sites using biopsied tumor samples. Our analysis can be most similarly compared to an approach used at the University of Iowa Hospital that has mapped out IHC tests on biopsied tumor samples with a classification model for four NET subtypes. Our top model, the SVM-RB model, with accuracy = 93.1%, sensitivity = 93.8%, and specificity = 92.3% on the held-out test set, performs similarly to the clinical IHC classification model, having a 90% sensitivity for sbNETs and around 90% sensitivity for pNETs, outperforming a reduced “community” model using commonly available stains [15,30]. Another notable approach uses quantitative polymerase chain reaction (qPCR) to assess the expression of 4 genes for differentiating pNETS and sbNETs and achieved an accuracy = 94% [15]. Our top model demonstrates similar or better performance compared to other multi-biomarker analyses referenced. It is worth underscoring that there are limitations observed in these studies, including issues related to sample size and study design, which raise concerns regarding the generality of the model and the practicality of widespread clinical implementation [13,15,21,22,25]. Our SVM-RB model comes with the limitation that is inherent with most machine learning models. In the context of clinical application, the model would need to be encapsulated within a deployable software package. If pursued, the objective would be to reduce the testing to only two or three of the most important variables or probes for model performance. This would include the *ERBB2*, *CKS1B*, and *CCNE1* FISH probes identified by PVI. The utilization of only two probes is warranted, primarily because the majority of samples in our dataset underwent two FISH probe tests, and we can expect similar performances. However, it is worth noting that additional tests could increase confidence in the metrics provided to the model for prediction. While having a deployable model is not uncommon, it does decrease its practicality. This is the rationale for optimizing the DT models, an approach that allows for direct clinical application due to the transparency of decisions. 

Our best training and testing performance for a DT model was achieved with KNN-imputed data, followed by a transformation to single-gene biomarker variables using biologically defined thresholds to denote a loss, normal, or gain status for standard FISH test results. Across all other models evaluated, a decrease in performance was observed with this transformation, consistent with expectation when converting continuous variables into categorical variables. With the DT models, the reduction in the number of variables through the incorporation of biological thresholds allowed for better performance and increased interpretability of the models in contrast to the tree fitted with the original continuous variables. Through our manual Monte Carlo CV with ten random 80/20 dataset splits, DT models generated tree plots with a range in complexity. Three trees featured only a single split, with two splitting on the *ERBB2* variable and one splitting on the *CDKN2A* variable. Training Set 6 and Training Set 9 yielded the most complex tree models, having 12 and 10 splits, respectively. The DT model produced from Training Set 9 exhibited the poorest performance on the held-out test set. This is expected, as the increase in splits leads to overfitting of the training data and a decrease in performance on newly presented data. On average, our DT models were not overfitting the training data and had better performance on the held-out test sets. 

In a clinical context, it is imperative to leverage all available data to inform decision-making for new samples. This is the rationale for constructing a DT model on the complete dataset. Our DT model trained on the entire dataset achieves an accuracy of 79.2%, classifying sbNET and pNET samples with consistent 10-fold CV performance across metrics. The limitation is that this model has not undergone testing on an independent held-out test set. Its performance is consistent with the mean performance observed in our manual Monte Carlo CV DT analysis and the consensus tree performance. These analyses are validation for the full dataset DT model, and it is expected to maintain its performance. In general, decision splits and PVI were consistent across all datasets and DT models constructed. The *ERBB2* variable was the most important variable for PVI and the primary split for the full dataset model, providing evidence for prioritization as a clinical FISH test. The *MET* variable was consistently included in the top variables for PVI across DT models. This can be attributed to being tested on the greatest number of samples compared to all other FISH probes and is important to model performance through the imputation of missing values. The *SMAD4* and *CDKN2A* are the other two variables included in the full dataset model and should be prioritized. The University of Iowa IHC model [15] outperforms our full dataset DT model when comparing sbNET sensitivity and specificity and pNET specificity, but our model has greater pNET sensitivity. The final full dataset model included, with accuracies at the terminal nodes, provides probabilities for assessing and diagnosing small bowel or pancreas primary sites of new NET samples, based on *ERBB2*, *SMAD4*, and *CDKN2A* FISH probes test results. 

A closer examination of the 25 samples predominately misclassified by our ten DT models reveals that 16 of the 25 samples are pNETs misclassified as sbNETs. The reason for the misclassification is due to the initial split in the *ERBB2* variable for most models, including the full dataset model, where these samples are pNETs but do not have a copy number gain for *ERBB2*. As for the other samples, no trends for misclassification have been identified. However, when selecting out these 25 samples and subjecting them to a DT algorithm for classification by themselves, an initial split on the *SMAD4* variable would correctly classify 21 of the 25 samples (84%). The challenge lies in identifying these samples in advance, where we are currently lacking the necessary link or variable(s) to do so. This is a potential area for further investigation for expansion of this analysis and our models. 

One of the main strengths of this study is the robust ML pipeline and analysis. Our analysis produces actionable results from real-world clinical data. Data gathered from clinical samples and electronic health records are not always complete across all samples. An integral component of our pipeline is the assessment of imputation strategies to address the gaps in the data, enhancing the performance of our analysis. All imputation strategies and dataset transformations, in conjunction with hyperparameter tuning, were implemented inside a stratified 10-fold cross-validation at each fold. This ensures a more robust estimation of model performance, eliminating any biases and data leakage from the validation sets. The initial step in our pipeline, prior to any data assessment or manipulation, is an 80/20 split of the entire dataset, altogether estimating performance on validation sets and assessing true performance on a held-out test set. When evaluating datasets and model parameters, selection of the model was based on the brier score. Preliminary analysis, indicating models selected based on other metrics, yielded weaker results when assessed on the held-out test set. In assessing overall model performance, we considered both training and test set performance, with a priority on the brier score, accuracy, and AUROC curve metrics. We examined multiple well-established and validated ML models previously demonstrated to produce insightful results on biological datasets [18]. Top-performing models are not always transparent with how predictions are produced. We demonstrate how to include and examine variables with a more transparent DT model that yields probabilistic guidance on predictions. In our DT analysis, we expanded the approach to a manual Monte Carlo CV to examine the multiple models constructed and establish a consensus of important variables from the full dataset. This method ensures that our initial random dataset 80/20 split and DT model were not presenting misleading results, and top results could be reproduced with additional splits. The PVI assessment for each model is an important element for the overall goal of the study, and resulting metrics can be compared across models and are not model specific. Ultimately, PVI and the variables utilized in the DT models allowed for a consensus for FISH probe prioritization. Our ML pipeline offers a robust framework for future analyses investigating similar problems. 

One limitation of this study is the moderate sample size of the dataset, which is partly attributed to the rarity of NETs, making it challenging to gather larger cohorts. Another limitation is the presence of missing values in our dataset. Although overcome by using imputation strategies, observed FISH results for all variables and all samples would yield more accurate overall results. Future studies aim to broaden the scope of biomarkers being investigated to include other important biomarkers shown to have predictive value, such as peripheral blood-based biomarkers and IHC [15,21,22,25,30]. Furthermore, we plan to extend our predictive models to encompass multiple primary sites, in addition to addressing both pNETs and sbNETs. 

In conclusion, we sought to prioritize FISH probes for diagnosing pancreas and small bowel primary sites for NETs using a multi-biomarker approach with ML techniques. The models can correctly classify pNETs and sbNETs with the widely used and relatively low-cost FISH tests. Results from our DT models and variable importance analysis offer probabilistic guidance for FISH testing, emphasizing the prioritization of *ERBB2*, *SMAD4*, and *CDKN2A* FISH assays in diagnosing NET primary sites.

## 4. Materials and Methods

Patients were recruited from the University of Iowa Hospital and Clinics. Patients were made aware of the present study and consented to participate. In total, 144 patients participated, consisting of 85 sbNETs diagnoses and 59 pNETs diagnoses. The FISH data used in this study was generated for validation of a discovery analysis for FISH probe association with NET primary sites. FISH probes were tested individually for statistical significance in differentiating NET subtypes and replication of initial findings. The sample size for each FISH probe and samples selected were guided by power analyses and tissue availability, resulting in an incomplete dataset with missing data for the present study [17]. Collection of samples, construction of tissue microarrays, and both chromosomal microarray and FISH testing have been described previously [17].

A total of 8 gene biomarker probes were assessed during FISH testing, including CDC28 protein kinase regulatory subunit 1B (*CKS1B*, 1q21.3), Fibroblast Growth Factor Receptor 3 (*FGFR3*, 14p16.3), Colony Stimulating Factor 1 Receptor (*CSF1R*, 5q32), MET Proto-Oncogene, Receptor Tyrosine Kinase (*MET*, 7q31.2), Cyclin Dependent Kinase Inhibitor 2A (*CDKN2A*, 9p21.3), Erb-B2 Receptor Tyrosine Kinase 2 (*ERBB2*, 17q12), SMAD Family Member 4 (*SMAD4*, 18q21.2), and Cyclin E1 (*CCNE1*, 19q12). Each FISH test consists of three metrics, percentage of nuclei exhibiting a loss of signal, a gain of signal, or no change in signal (normal/wild-type), with ranges of 0–100 (averaged metric from test done in triplicate). Not all gene biomarker probes were assessed with FISH testing for all samples. FISH test metrics were utilized as predictor variables for classification of patients with sbNETs or pNETs with machine learning models. Patients were diagnosed and labeled as either sbNET or pNET by a combination of standard care tests including imaging and immunohistochemistry tests.

Dataset curation, preprocessing, and machine learning analysis was performed on the University of Iowa’s high-performance computing cluster within the R programming language (version 4.1.3) [31]. To address multiple model testing and provide a measure against overfitting and detection of false positive patterns in the data, machine learning analyses used an 80/20 training-testing split, reserving 20% of the data as a held-out test set. The initial dataset consists of 25 variables: a loss, normal, and gain variable for each of the 8 gene biomarker FISH tests and the response variable. Preprocessing of the training dataset was implemented within a pipeline, with the recipes R package (version 1.0.5) allowing for each preprocessing step to be performed at each resampling fold for estimating predictive performance of models, and subsequently, with the held-out test set for assessing true predictive performance [32]. We employed imputation techniques to address the missing data as a preprocessing step, driven by the rationale that a complete dataset would enhance our analysis. The methods utilized were impute by mean, median, k-nearest neighbor (KNN), and bagged tree models (tree = 25) and assessed for effect on the performance of models. KNN imputation was implemented by identifying the five nearest samples (k = 5) based on Gower’s distance. These samples included both present and missing variables of the sample being imputed, enabling the mean of the five samples to be used for the missing variable(s). After imputation of missing data, predictor variables were transformed into single categorical biomarker variables with 1 denoting a loss, 2 denoting a normal, and 3 denoting a gain for the copy number of a gene biomarker. Loss, normal, and gain values were determined by thresholds set on the original variables. The threshold was set based on the percentage of nuclei that exhibited the change which yielded cut-off thresholds of greater than 28.3% for a loss, greater than 68.3% for a normal, and greater than 15.2% for a gain value. These thresholds were validated for these specific probes on these tissue sample types previously [17]. Transformed biomarker variables were assigned the value for the variable of the highest exceeding value when multiple variables breached the thresholds and were assigned normal/wild-type (2) when no variables passed the thresholds. Multiple datasets were constructed and tested with various imputation and transformation strategies to find an optimally performing dataset.

Models were trained and performance-assessed for each of the following datasets: original variables without imputation, original variables with imputation, and transformed variables with imputation. The models assessed were decision tree (DT), random forest (RF), extreme gradient boosted tree (XGBTree), lasso, elasticnet, logistic regression (log reg), naïve bayes (NB), support vector machine with radial basis kernel function (SVM-RB), support vector machine with linear kernel function (SVM-L), and support vector machine with polynomial kernel function (SVM-P). Model hyperparameter tuning was completed using a grid search with stratified 10-fold cross-validation (CV) for model and parameter selection, as well as to estimate predictive performance recorded as mean metrics. Parameter values for each model are listed in Appendix A (Appendix A). Calculated training and testing performance metrics include brier score, accuracy, Cohen’s kappa, area under the receiver operating characteristic (AUROC) curve, sensitivity, and specificity. Selection of best performing model was based on brier score, due to brier score being a proper scoring function, with measuring of the accuracy based on prediction probabilities. Best performing models were used for prediction of samples in the held-out test sets. For each model, relative permutation-based variable importance (PVI, samples = 25) was calculated and reported for each initial input variable. Compared to other machine learning models, decision tree models offer high interpretability, providing clear insights into predictions. For this reason, decision tree models were assessed on a total of ten random 80/20 training testing splits for manual Monte Carlo CV of the transformed variable dataset for predictive performance, PVI, and variable splits included in each model. A consensus tree was built from the mean prediction probabilities of each sample across all ten models using the full dataset and assessed for predictive performance. A decision tree model was fit to the complete transformed variable dataset and assessed for predictive performance, PVI, and variable splits. Machine learning analysis including models trained, performance calculations, tree plots, and PVI plots, and ROC curves were performed using the MachineShop R package (version 3.6.2) [33].

## Figures and Tables

**Figure 1 ijms-24-17401-f001:**
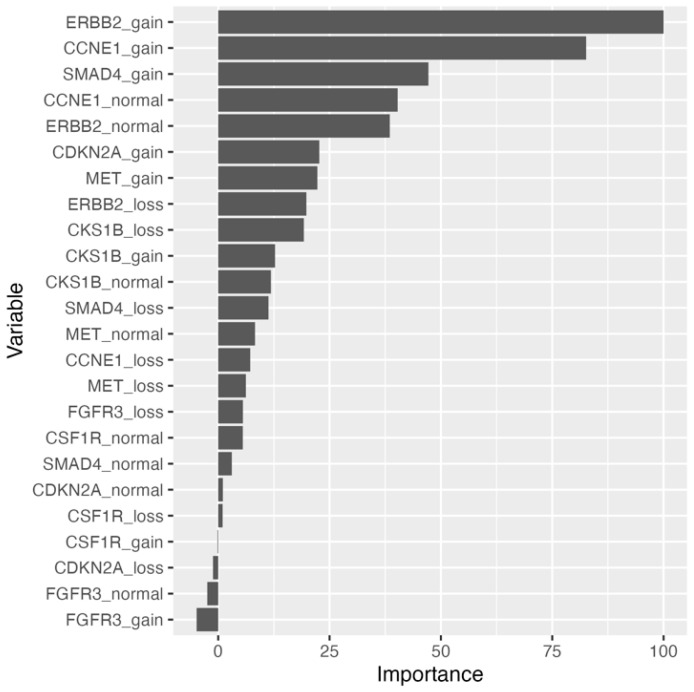
Permutation-based variable importance for naïve bayes model trained on raw dataset. All variables included in the model are assessed. Permutations per variable were performed 25 times and performance is assessed based on mean brier score.

**Figure 2 ijms-24-17401-f002:**
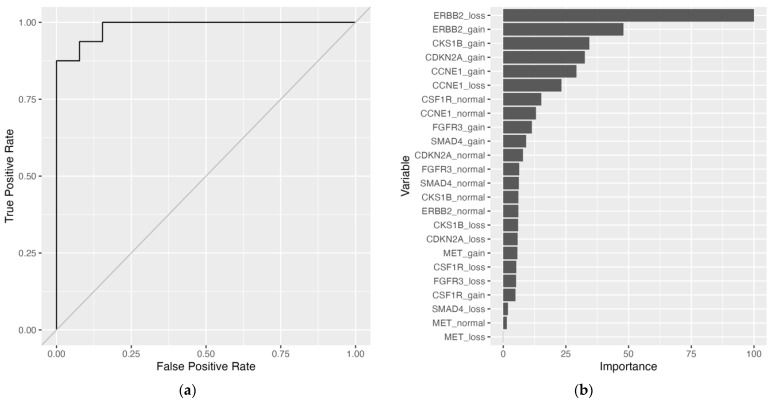
Support vector machine with radial basis kernel function (SVM-RB) model trained on median-imputed dataset. (**a**) Receiver operating characteristic (ROC) curve. Area under the receiver operating characteristic (AUROC) curve = 0.986. (**b**) Permutation-based variable importance. All variables included in the model are assessed. Permutations per variable were performed 25 times and performance is assessed based on mean brier score.

**Figure 3 ijms-24-17401-f003:**
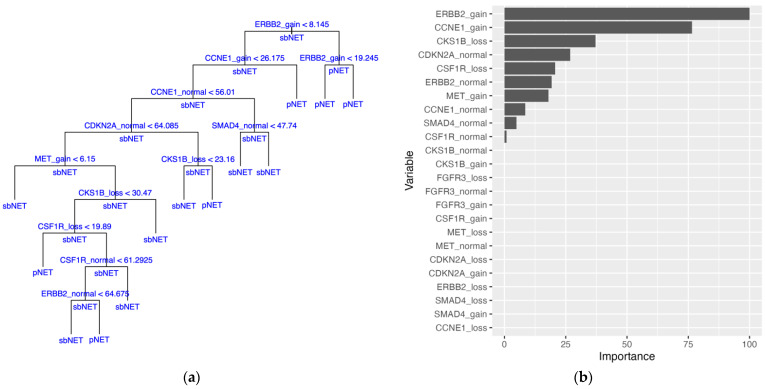
Decision tree (DT) model trained on median-imputed data. (**a**) DT model with internal nodes shown with variables being split with thresholds. Samples with a value for a variable less than the threshold are split down the left branch and samples with a value greater than the threshold are split down the right branch. Terminal nodes show the final classification of the samples with either pancreatic neuroendocrine tumor (pNET) or small bowel neuroendocrine tumor (sbNET). Classification of sample subsets at internal nodes are shown under horizontal lines. (**b**) Permutation-based variable importance. All variables included in the model are assessed. Permutations per variable were performed 25 times and performance is assessed based on mean brier score.

**Figure 4 ijms-24-17401-f004:**
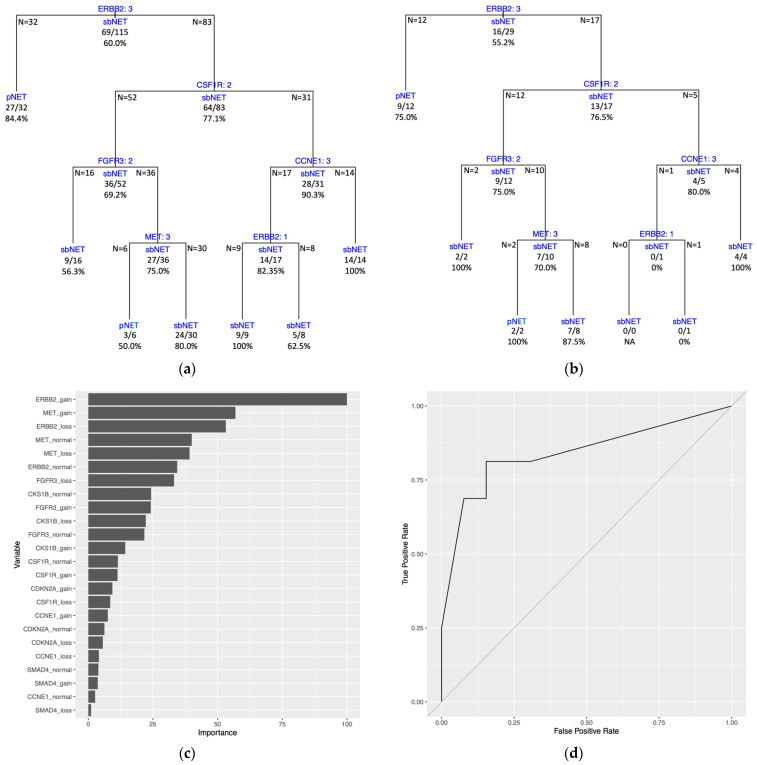
DT model trained on KNN-imputed data with a transformation of the data to single categorical variables for each biomarker. Training and test sets are from initial random 80/20 dataset split (Training and Test Set 1). (**a**) Training set DT model with internal nodes shown with variables being split with splitting criteria of 1 = loss, 2 = normal, and 3 = gain for number of copies at the genomic regions. Samples categorized with a variable value equal to the splitting criteria are split down the left branch and those with a variable value different than the splitting criteria are split down the right branch. Terminal nodes show the final classification of the samples with either pNET or sbNET. Classification of sample subsets at internal nodes are shown under horizontal lines and sample numbers of subsets from the split are displayed at the edges of the horizontal lines. Each node lists correctly classified samples as a fraction and percent accuracy. Tree accuracy = 79.1% (91/115). (**b**) Test set DT model plot with same design as (**a**) with test set data. Tree accuracy = 82.8% (24/29). (**c**) Permutation-based variable importance. All variables included in the model are assessed. Permutations per variable were performed 25 times and performance is assessed based on mean brier score. (**d**) ROC curve. AUROC curve = 0.841.

**Figure 5 ijms-24-17401-f005:**
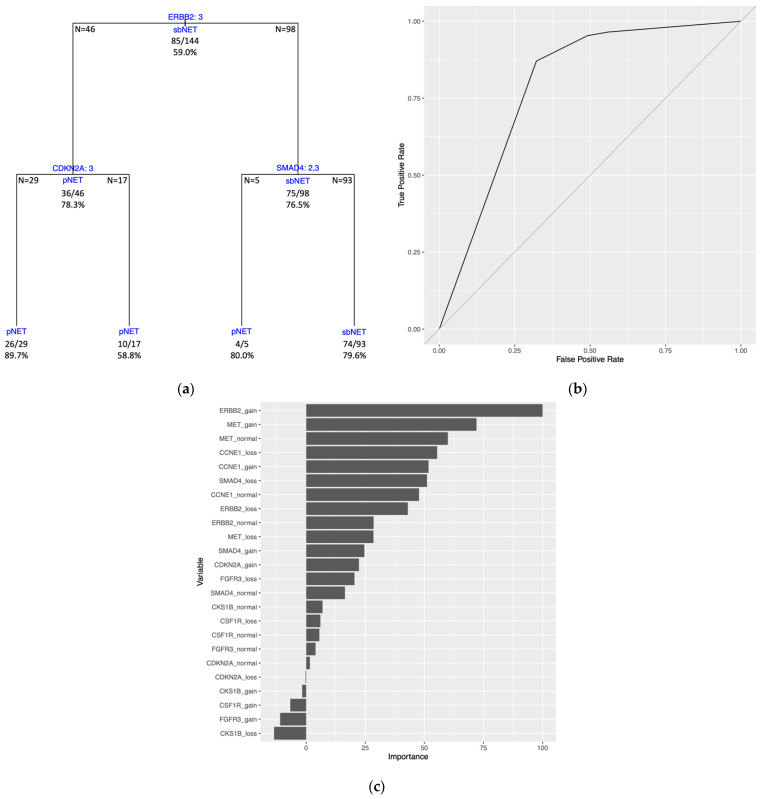
Full dataset DT model trained on all samples with KNN-imputed data with a transformation of the data to single categorical variables for each biomarker. (**a**) DT model with internal nodes shown with variables being split with splitting criteria of 1 = loss, 2 = normal, and 3 = gain for number of copies at the genomic regions. Samples categorized with a value for a variable equal to the splitting criteria are split down the left branch and samples with a value different than the splitting criteria are split down the right branch. Terminal nodes show the final classification of the samples with either pNET or sbNET. Classification of sample subsets at internal nodes are shown under horizontal lines and sample numbers of subsets from the split are displayed at the edges of the horizontal lines. Each node lists correctly classified samples as a fraction and percent accuracy. Tree accuracy = 79.2% (114/144). (**b**) ROC curve. AUROC curve = 0.793. (**c**) Permutation-based variable importance. All variables included in the model are assessed. Permutations per variable were performed 25 times and performance is assessed based on mean brier score.

**Table 1 ijms-24-17401-t001:** Distribution of the number of fluorescence in situ hybridization (FISH) tests performed on samples.

Number of FISH Tests	Number of Samples
1	5
2	68
3	13
4	41
5	1
6	10
7	6
8	0

**Table 2 ijms-24-17401-t002:** Distribution of FISH tests performed on samples.

Biomarker	Number of Samples
*CKS1B*	56
*FGFR3*	57
*CSF1R*	54
*MET*	93
*CDKN2A*	36
*ERBB2*	73
*SMAD4*	41
*CCNE1*	41

**Table 3 ijms-24-17401-t003:** Mean 10-fold cross-validation (CV) Training set performance and held-out test set performance of top models from raw and imputed datasets.

**Training Performance**												
	**Raw Data**		**Mean**		**Median**			**KNN**			**Bagged Trees**	
Top Models	NB	XGBTree	SVM-RB	Log Reg	SVM-RB	Log Reg	DT	SVM-RB	Log Reg	NB	SVM-RB	Log Reg
Brier Score	0.156	0.157	0.117	0.172	0.115	0.251	0.175	0.115	0.228	0.176	0.119	0.190
Accuracy	83.5%	79.9%	86.2%	78.2%	85.4%	74.9%	75.5%	86.3%	72.1%	81.9%	83.5%	78.5%
AUROC curve	0.869	0.861	0.924	0.839	0.919	0.739	0.826	0.908	0.756	0.877	0.909	0.803
**Test Performance**												
	**Raw Data**		**Mean**		**Median**			**KNN**			**Bagged Trees**	
Top Models	NB	XGBTree	SVM-RB	Log Reg	SVM-RB	Log Reg	DT	SVM-RB	Log Reg	NB	SVM-RB	Log Reg
Brier Score	0.117	0.166	0.069	0.069	0.066	0.066	0.183	0.077	0.077	0.105	0.068	0.068
Accuracy	86.2%	79.3%	93.1%	93.1%	93.1%	93.1%	72.4%	89.7%	89.7%	89.7%	93.1%	93.1%
AUROC curve	0.928	0.856	0.981	0.981	0.986	0.986	0.820	0.976	0.976	0.952	0.962	0.962

**Table 4 ijms-24-17401-t004:** Mean 10-fold CV Training set performance and held-out test set performance of top models from imputed datasets with a transformation of the data to single categorical variables for each biomarker.

**Training Performance**								
	**Mean**	**Median**		**KNN**			**Bagged Trees**	
Top Models	RF	RF	SVM-RB	DT	RF	SVM-RB	RF	SVM-P
Brier Score	0.174	0.187	0.168	0.180	0.176	0.168	0.211	0.179
Accuracy	77.2%	76.6%	77.4%	78.1%	74.0%	79.1%	73.1%	77.5%
AUROC curve	0.797	0.766	0.843	0.781	0.806	0.786	0.726	0.759
**Test Performance**								
	**Mean**	**Median**		**KNN**			**Bagged Trees**	
Top Models	RF	RF	SVM-RB	DT	RF	SVM-RB	RF	SVM-P
Brier Score	0.147	0.133	0.158	0.157	0.154	0.169	0.132	0.200
Accuracy	75.9%	79.3%	79.3%	82.8%	79.3%	79.3%	79.3%	72.4%
AUROC curve	0.885	0.930	0.916	0.841	0.873	0.817	0.916	0.829

**Table 5 ijms-24-17401-t005:** Summary statistics for mean 10-fold CV Training set performance and held-out test set performance of top ten DT models from manual Monte Carlo CV with ten random 80/20 dataset splits. DT models are trained on KNN-imputed data with a transformation of the data to single categorical variables for each biomarker.

**Training Performance**						
	**Brier Score**	**Accuracy**	**Kappa**	**AUROC Curve**	**Sensitivity**	**Specificity**
Minimum	0.161	63.5%	0.250	0.696	0.662	0.570
Maximum	0.228	80.4%	0.600	0.827	0.895	0.760
Mean (SD)	0.185 (0.020)	75.9% (0.050)	0.486 (0.099)	0.764 (0.047)	0.846 (0.068)	0.633 (0.056)
**Test Performance**						
	**Brier Score**	**Accuracy**	**Kappa**	**AUROC Curve**	**Sensitivity**	**Specificity**
Minimum	0.104	69.0%	0.393	0.710	0.727	0.546
Maximum	0.252	89.7%	0.784	0.909	1.000	0.900
Mean (SD)	0.156 (0.043)	81.4% (0.068)	0.598 (0.127)	0.816 (0.072)	0.864 (0.090)	0.732 (0.140)

## Data Availability

FISH data analyzed during the study are available from the corresponding author on request. R libraries used by this study are publicly available at the CRAN website: https://cran.r-project.org (accessed on 15 May 2023). Software used for the study can be found at the GitHub website: https://github.com/lpietan/NET-ML (accessed on 20 October 2023).

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
