# Peer review of "Prioritization of Fluorescence In Situ Hybridization (FISH) Probes for Differentiating Primary Sites of Neuroendocrine Tumors with Machine Learning"

_ijms, 2023, doi:10.3390/ijms242417401_

Round 1
Reviewer 1 Report
Comments and Suggestions for Authors
The manuscript “Prioritization of Fluorescence in Situ Hybridization (FISH) 2 Probes for Differentiating Primary Sites of Neuroendocrine 3 Tumors with Machine Learning” by Lucas Pietan et al reports results from machine learning method on classification of patients as with pancreatic NETs (pNETs) or small bowel NETs (sbNETs) using 8 FISH probes data. Ten methods were applied to the data set with 144 patients. Although relative high accuracy rate and AUROC were reported, The estimated Kappa is low for the chosen focus DT method.
Overall, the manuscript is well prepared, here are some concerns.
1. Higher rate of missing data, although different methods were used to impute the missing data, some of those methods may not be helpful to increase predictive accuracy (Median, mean imputation).
2. For the method of KNN to impute the missing, how to choose which variables in use to impute the missing?
3. The correlation matrix of the 8 FISH probes raw data may help to determine which method to choose for imputing missing.
4. To report the predictive accuracy, it should be based on left out test data, not the left-out subset in the training set in CV, as that was used for model selection, over estimate the true predictive accuracy.
5. Some reference (ref 17) is not available to reviewer.
Reviewer 2 Report
Comments and Suggestions for Authors
The author analyzed the significance of FISH for ERBB2, CDKN2A, and SMAD4 in determination of primary site of neuroendocrine tumors.
This article sounds interesting, and can provide important information in the field of research for neuroendocrine tumors.
Discussion of genetic background of three genes in neuroendocrine tumors of pancreas and small intestine must be added.
As authors mentioned, FISH is one of useful methods, however, immunohistochemical analysis is more convenient method. Is there useful immunohistochemical marker detecting primary site of neuroendocrine tumors?
